# Identification and Quantification of β-Sitosterol β-d-Glucoside of an Ethanolic Extract Obtained by Microwave-Assisted Extraction from *Agave angustifolia* Haw

**DOI:** 10.3390/molecules24213926

**Published:** 2019-10-31

**Authors:** Herminia López-Salazar, Brenda Hildeliza Camacho-Díaz, Sandra Victoria Ávila-Reyes, Ma Dolores Pérez-García, Manases González- Cortazar, Martha L. Arenas Ocampo, Antonio R. Jiménez-Aparicio

**Affiliations:** 1Centro de Desarrollo de Productos Bióticos, Instituto Politécnico Nacional, P.O. Box 24, Yautepec 62730, Morelos, Mexico; herminia784@gmail.com (H.L.-S.); bhcamachod@gmail.com (B.H.C.-D.); sandra_victory@yahoo.com (S.V.Á.-R.); mlarenas@ipn.mx (M.L.A.O.); 2Centro de Investigación Biomédica del Sur, Instituto Mexicano del Seguro Social ((CIBIS-IMSS), Xochitepec 62790, Morelos Mexico; lola_as@yahoo.com (M.D.P.-G.); gmanases@hotmail.com (M.G.-C.)

**Keywords:** *Agave angustifolia* Haw, β-sitosterol β-d-glucoside, MAE

## Abstract

β-sitosterol β-d-glucoside (BSSG) was extracted from “piña” of the *Agave angustifolia* Haw plant by microwave-assisted extraction (MAE) with a KOH solution such as a catalyst and a conventional maceration method to determine the best technique in terms of yield, extraction time, and recovery. The quantification and characterization of BSSG were done by high-performance thin layer chromatography (HPTLC), Fourier-transform infrared spectroscopy (FT-IR), and high-performance liquid chromatography−electrospray ionization−mass spectrometry (HPLC-ESI-MS). With an extraction time of 5 s by MAE, a higher amount of BSSG (124.76 mg of β-sitosterol β-d-glucoside/g dry weight of the extract) than those for MAE extraction times of 10 and 15 s (106.19 and 103.97 mg/g dry weight respectively) was shown. The quantification of BSSG in the extract obtained by 48 h of conventional maceration was about 4–5 times less (26.67 mg/g dry weight of the extract) than the yields reached by the MAE treatments. MAE achieved the highest amount of BSSG, in the shortest extraction time while preserving the integrity of the compound’s structure.

## 1. Introduction

The genus Agave belongs to the Agavaceae family of which 200 species have been documented; 150 (75%) are distributed in Mexico and 116 (58%) are endemic [1,2]. These agaves were a source of carbohydrates (before the corn crop was established) for western Mexico and the southwestern United States of America [3]. Furthermore, the agaves have had and continue to have great social, cultural, ecological, environmental, and medicinal importance. *A. angustifolia* is an example of this family, which has been used in various ways such as the production of alcoholic beverages such as the “bacanora” of Sonora and the “mezcal” in southern Mexico. Its leaves are used to cook roasted lamb and flowers in various dishes. In some parts, the flowers and leaves are used as fodder. Likewise, its fibers are extracted to make ropes, baskets, clothes, sandals, brushes, etc. In traditional Mexican medicine, it has been widely used by the indigenous people to heal different ailments such as dehydration, scurvy, blows, wounds, indigestion, “reumas,” toothache, stings of poisonous animals, and to heal livestock diseases. In addition, it is used as a remedy for the sprains of bones of people and animals [4].

A large range of secondary metabolites have been reported from different Agave species, such as steroidal sapogenins and saponins, sterols [5,6,7], fructans [8], flavonoids [9,10,11], homoisoflavonoids [12], phenolic acids [10], tannins, volatile coumarins [9], long-chain alkanes, fatty acids, and alcohols [12,13]. Evidence of pharmacological activity has been reported for this genus, where immunomodulatory, anti-inflammatory, cytotoxic, and antiparasitic activity are some of the most important [6,7,14,15].

An example of these biological activities was reported in a plant of this genus, *A. angustifolia* [6], where an immunomodulatory effect of the acetonic extract was found by the maceration method. It has been demonstrated that the compounds responsible for this activity were 3-*O*-[(6′-*O*-palmitoyl) -β-d-glucopyranosyl] sitosterol, stigmasterol, and β-sitosteryl glucoside. The β-sitosterol β-d-glucoside (BSSG) and its free phytosterol, known as β-sitosterol, are phytosterols that have been described as bioactive molecules, which provide anti-inflammatory activity [16], immunomodulation activity [17], and protection against certain cancers [18], as well as reduce cholesterol [19] and control blood glucose [20]. Therefore, they can be suitable candidates for the treatment of different diseases.

It is important to mention that maceration has been a traditional extraction method of phytosterols. This extraction technique involves soaking plant materials (coarse or powdered) in a stoppered container with a solvent at room temperature for a period of 72 h, sometimes assisted with frequent agitation. Although this method is easy and simple to perform, the prolonged extraction time is inconvenient and requires a large quantity of toxic solvents [21].

In recent years, novel extraction techniques have been developed such as ultrasound-assisted extraction (UAE), pressure-assisted extraction (PAE), and microwave-assisted extraction (MAE); these techniques have shown advantages over conventional extraction methods, such as greater yield, shorter time, and less required solvent [22].

Microwave extraction technology is recognized as an environmentally friendly method for the extraction of medicinal and aromatic plants compared to conventional techniques, by providing a shorter extraction time, using environmentally friendly solvents, reducing the overall consumption of solvents and energy during the process, and developing a selective extraction with an increase in the yield. In addition to having a good reproducibility as an automated technique; it uses minimum manipulation of the sample for the extraction process [23].

MAE uses microwave energy to accelerate the partition of analytes from the sample matrix into the solvent. Microwaves are electromagnetic waves that consist of an electric field and a magnetic field, which oscillate perpendicularly to each other at frequencies from 0.3 to 300 GHz. Microwave radiation interacts with the dipoles of polar and polarizable materials and ions, which causes heating of the materials dependent on the materials’ dielectric properties and presentation geometry. Equally important, dipole rotation of the molecules induced by microwave electromagnetic disrupts hydrogen bonding, improves the migration of dissolved ions, and promotes solvent penetration into the matrix [24]. MAE can be considered a selective method that favors polar molecules and solvents with a high dielectric constant [25].

MAE has been used for the extraction of several classes of compounds, such as essential oils [26], terpenes [27,28], flavonoids, phenols [27], alkaloids [29], and glucoside [30] from natural-plant resources. 

The aim of this study was to evaluate the extraction of BSSG from *A. angustifolia* by the MAE and maceration methods in terms of extraction time and quantification using environmentally friendly solvents.

## 2. Results and Discussion

### 2.1. High-Performance Thin Layer Chromatographic (HPTLC) Analysis

The first step in this research work was to determine the usefulness of applying a solution with KOH to the extracts obtained by microwaves, in order to improve the extraction of BSSG. To assess this objective, a qualitative HPTLC plate Figure 1 was prepared using a mobile phase of toluene:ethyl acetate:formic acid in the ratio (5:5:0.7 *v*/*v*/*v*). The derivatized plate with the Komarovsky reagent caused the bands corresponding to terpenes to have a purple color. In the qualitative HPTLC plate, we can appreciate the greater purple intensity of the band corresponding to the BSSG in the extracts to which the KOH solution was applied, compared to the extracts without its application. This was the criterion taken to perform the quantification of BSSG in the extracts with KOH solution, using microwaves as the extraction method.

The identification of BSSG was determined in the experimental extract samples by obtaining the same value of the retention factor (rf), 0.230 +/− 0.020, of the BSSG reference standard. This value of the rf is the same as that reported by Jirge et al. in 2011 [31], who reported a rf of 0.21 in a mobile phase consisting of chloroform:methanol (8:2 *v/v*). 

Figure 2 shows a quantitative HPTLC plate, in which adequate separation of the line of the application of BSSG standard was observed. Additionally, the presence of BSSG was confirmed again in the ethanol extracts obtained by MAE (5, 10, and 15 s with KOH solution) and maceration (48 h) from *A. angustifolia*. 

They performed the standardization of four formulations for their content of BSSG, using HPTLC. In order to carry out the comparison between the extracts and the standard solution, integration was performed to separate the zone of the chromatogram containing the BSSG in the extracts.

Densitograms of the BSSG standard and the BSSG in the extracts are shown in Figure 3, which demonstrate the similarity of the rf of the standard and the extracts of interest.

After the plate was derivatized, a densitometric scan was performed, which allowed the band to be located in an automated way by scanning at different wavelengths from λ = 200 to 600 nm. Figure 4 shows the comparison and the spectral similarity obtained from the BSSG standard with the different extracts of interest (ethanolic extracts of MAE at 5, 10, and 15 s with KOH solution and the extract by maceration of 48 h). The height of the peaks increased proportionally with the increase in the concentration of BSSG in the sample, with the highest absorption peak at λ = 540 nm for the BSSG.

The results of the quantification of BSSG are shown in Table 1. The extraction time of 5 s by MAE with KOH solution yielded the highest amount of BSSG, achieving 124.76 mg of BSSG/g dry weight of the extract, compared to other times of MAE of 10 s (106.19 mg/g dry weight) and 15 s (103.97 mg/g dry weight) both with KOH solution. Contact time with the extraction system may compromise the chemical structure of the metabolite, resulting in a greater degradation effect on polar compounds due to the microwave energy effect [32]. The quantification of BSSG in the extract obtained by 48 h maceration was 26.67 mg/g of dry weight of extract, which was substantially lower than the amount achieved in the three extraction times by MAE with a shorter extraction time.

To understand these results, it is important to remember the mechanism by which MAE uses microwave energy to facilitate the partition of analytes. First, the rapid rise in temperature and internal pressure increase generated by the microwave radiation pushes the cell wall from the inside, stretching and ultimately rupturing it, releasing compounds. This process is caused by microwave radiation, which interacts with dipoles of the polar and polarizable materials. Dipole rotation of the molecules produced by microwave electromagnetic waves cuts-off hydrogen bonding of the cellulose (which is the main constituent of the cell wall of the plant), which improves the migration of dissolved ions and stimulates solvent penetration into the vegetable matrix [25].

However, when we added the KOH as a catalyst, we caused more destructive effects on the vegetal matrix. This caused the BSSG to dilute and dissolve in the solvent in a faster way.

In this way, we can confirm with these results that the extraction of BSSG by MAE was favored by the addition of KOH by causing a greater decomposition of the cell wall, in a short extraction time, thus avoiding the degradation of the compound when exposed to a prolonged extraction time. This was probably the reason why a greater amount of BSSG was obtained in the extraction time of 5 s with KOH solution than 10 and 15 s with KOH solution.

The scientific literature reports that one of the benefits of MAE is the greater amount of the compound of interest obtained, compared to conventional extraction methods, such as maceration. The maceration method consists of the penetration of the solvent into the cell, causing dehydration or rupture of cell membranes [21].

In this work, the large amount of BSSG obtained by MAE compared to the method of maceration, is, in itself, due to the same MAE technique, which also benefits from the addition of the KOH solution.

The calibration curve of BSSG was linear in the range between 100 and 1000 ng/band, with a regression coefficient of R = 99.63% and a regression equation, y = 2.078 × 10^−10^ X+ 4.079 × 10^−3^.

The quantification results obtained can be compared to a research work, in which extraction by hexane maceration of the aerial parts of the *Sisymbrium irio* plant was realized, reporting a quantification of BSSG of 0.00210 mg/g dry weight [33], less than the amount obtained in our work by the MAE method as per the maceration technique. Our results are the first reported work in the extraction of BSSG by the MAE method, and they are similar to others that compared different conventional extraction methods with the MAE method. They concluded that the MAE method is a good and safe technique for the extraction of active metabolites from plants. There are different factors that influence the success of the MAE method, such as the selection of a suitable solvent that depends on the solubility of the compounds of interest, the penetration of the solvent, the interaction with the matrix of the plant material, and, furthermore, the material’s dielectric constant. There are organic solvents, such as ethanol, methanol, and acetone, that are also effective for application in the MAE. We know that water is a nontoxic and cheaper solvent, but it has some disadvantages such as being a good medium for the development of mold and bacteria; it can cause the breakdown of plant metabolites; besides, the evaporation of the extracts requires the application of high temperatures, which can favor the degradation of many of the compounds present in an extract [32]. Ethanol is usually the most used solvent because it is a good microwave absorbent, together with the advantage that is appropriate to extract different active compounds from plants [27,34]. Other investigations have reported the extraction and identification of BSSG, but they used toxic solvents such as petroleum ether and chloroform [35].

For all these reasons, the MAE method is known as an alternative ecological extraction technology, and a good option among thermal extraction techniques. This is due to its unique effective mechanism as a noncontact energy source to produce heat in the extraction matrix for effective heating, faster thermal energy transfer, less thermal degradation, higher extraction selectivity, a faster start of the extraction process (automated), and a higher yield in a shorter time, compared to conventional extraction methods [26].

### 2.2. Identification of the Functional Groups of BSSG by FT-IR

To carry out the FT-IR analysis, samples of the MAE extractions were taken at 5, 10, and 15 s with KOH solution to compare the presence of the functional groups present in the BSSG FT-IR spectrum. Figure 5A,B (FTI-IR zoom) show that the BSSG FT-IR spectrum showed an absorption band (cm^−1^) at 3406 in the presence of OH stretch, 2932 (CH_2_) -CH), CH_2_ a 2868.14, unconjugated olefinic (C = C) in 1641, cyclic methylene groups (CH_2_) *n* in 1443, gem-dimethyl group (-CH (CH_3_) 2) in 1366, secondary alcohol (C-OH) in 1055; while the absorption bands in 801 and 600 are due to the presence of a group -C=C-, possibly of simple glycoside. It is evident that the functional group characteristics of BSSB were identified in the FT-IR spectrum extracts obtained by MAE of 5, 10, and 15 s with KOH solution. The FT-IR spectra also showed the presence of one C=C in the structure, characteristic of BSSG (Figure 5A) [36,37].

It is worth mentioning that the FT-IR spectrum of extracts showed a clear displacement of some absorbances such as the unconjugated olefinic (C=C) band, occurring from 1641 to cm^−1^, cyclic methylene groups (CH2)n band, passing from 1443 to cm^−1^, C-OH of secondary alcohol band, occurring from 1055 to cm^−1^, and -C=C- group band, which takes place from 801 to 600 cm^−1^, respectively. These displacements in these bands can be related to the probable hydrolysis of the compound, which is due to the application of KOH solution, shown more clearly in Figure 5B. This phenomenon is better observed in the extraction time of 15 s with KOH solution. It is necessary to mention that the degradation of sterols takes place at more than 150 °C, resulting in fragmented phytosterol molecules, volatile compounds, and oligomers [38]. In this research work, the maximum extraction temperature was 23 °C; for this reason, we believe that the possible degradation was caused by the addition of KOH solution and not by microwave energy. The scientific literature mentions that the microwaves used in the extraction of bioactive compounds are a source of heat without contact; for this reason, there is a decrease in thermal degradation [22]. Besides, it is necessary to mention the FT-IR spectra are of pure extracts, and for this reason, the bands present can be associated with other compounds, for example, the pronounced band at 1641 cm^−1^ could be for the presence of lignins, which are polymeric aromatic constituents in plant cell walls and are found in this spectral region [39]. Nowadays, different researches have reported the use of FT-IR as a complementary method for the identification of sterols from plants, animals, and algae [40,41,42]. This phenomenon is due to FT-IR, which presents some advantages such as being a fast technique, which provides a great deal of information with only one test and does not need reagents and pre-treatment of the sample. Besides, this technique uses a relatively small sample quantity that can be recovered after the analysis [43]. For this reason, FT-IR analysis is a great tool for the identification of the chemical nature of phytochemical compounds present in medicinal plants, because they can contribute with information of the functional groups responsible for their biological activity [40].

### 2.3. High-Performance Liquid Chromatography−Electrospray Ionization−Mass Spectrometry (HPLC-ESI-MS) Analysis

The criteria that were taken into account to choose the best extract by MAE, to continue with the HPLC-ESI-MS analysis, were the data obtained in the quantification performed by HPTLC (highest yield of BSSG) and the results obtained in the FT-IR analysis (greater integrity of functional groups); as a result, the extract obtained by MAE of 5 s with KOH solution was chosen to continue with the analysis previously mentioned.

Our compound of interest, BSSG, has a real molecular weight of 576, and the mass spectrum in the 5 s ethanolic extract obtained by MAE shows a peak at 599.42 *m/z* = [M + Na]^+^, which is due to an adduct of a Na^+^. According to the spectral data found in this research work, it was confirmed that the compound identified and quantified from an ethanol extract of *A. angustifolia* obtained by MAE of 5 s is BSSG (Figure 6) [36].

In all plant tissues, sterols exist in two forms: Free form (free sterols, FSs) and the conjugated form, which includes steryl esters (SEs), steryl glycosides (SGs), and acyl steryl glycosides (ASGs). These conjugated sterols are found ubiquitously in plants, but their relative contents are different between species and can change in response to developmental and environmental signals. The conjugated sterols have important functions in the plant, such as the SEs acting in homeostasis of the membrane sterols, and are also used as a reserve of sterols in some plant tissues. The SGs and ASGs are found in the plant plasma membrane (PM) where they are accumulated in microdomains (lipid rafts) known to regulate many important cellular processes. Besides, there are different studies that confirm the role of conjugated sterols in plant stress responses [44]. In general, the phytosterols have been described as bioactive compounds, because they have important pharmacological activities that supply different benefits to human health [45]. To simplify the process of phytosterol analysis, conjugated phytosterols are frequently converted to free phytosterols through acid hydrolysis and base saponification before quantification, making quantification easier because only the total sterol content is obtained and the SGs and ASGs are ignored by the research. This could be because polar conjugated sterols are not soluble in the nonpolar lipid phase and may not be included in the direct lipid extracts [46]. However, nowadays, there is enough research evidence that suggests the importance of biological functions of SGs. This act is due to the presence of the steroidal backbone as the aglycone part increases its solubility in nonpolar solvents such as chloroform, whereas the presence of glucose moiety makes it slightly soluble in polar solvents such as ethanol or methanol [47]. This characteristic provides an important advantage if we compare it with the FSs, and one example of this is that SGs contain more hydrophilic parts (glucose moiety); this part of these glucosides is being thought to effectively hinder the esterification of cholesterol, thus resulting in the inhibition of entry of cholesterol into the blood vessel; this medical activity has been reported [48].

Another important point to take into account in the medicinal activity of plant extracts is the importance of the use of pure extracts, because the components may interact and generate a synergistic effect, thus favoring a biological activity of interest, better than when the compounds are used in isolation [49].

## 3. Experimental Section

### 3.1. Extraction Techniques

#### 3.1.1. Plant Material

A 5 year old “piña” of the *A. angustifolia* plant of recorded biological identity was collected in Yautepec, Morelos, Mexico (18°49′33.3”N and 99°06′21.98″W; 1120 m above mean sea level).

#### 3.1.2. Sample Preparation

The “piña” of the *A. angustifolia* plant was fractionated to produce 32.4 kg of fresh material. After placing the plant material in an oven at a constant temperature of 40 °C for 48 h, the dry plant material was ground (INMIMEX Mill, M-200, Mexico City, Mexico) using a 200 mesh sieve (0.074 mm particle size), resulting in a final yield of 3965 kg.

#### 3.1.3. Innovation Extraction Technique: Microwave-Assisted Extraction (MAE)

MAE was carried out in a laboratory CEM Discover^®^ microwave oven (300 W at 2450 MHz equipment maximum power), Matthews, NC, USA. An open MAE system was used in this work, which operates at atmospheric conditions, and only the vessel was exposed to the microwave radiation.

The MAE conditions used were a constant radiation power of 300 W, a ratio of 20:1 mL/g of liquid/solid, 23 °C of extraction temperature maximum, different extraction times of 5, 10, and 15 s, besides the application of 1 mL of ethanolic solution of KOH 0.2 N. Afterward, all samples were filtered and concentrated at low temperature and reduced pressure on a rotary evaporator (Heildolph Laborota Model 4000, Schwabach, Germany).

It is necessary to mention that the application of an ethanolic solution of KOH 0.2 N was determined in preliminary experiments (qualitative HPTLC plate) to find its effect on BSSG extraction and its relation to microwave extraction. Three extraction times (5, 10, and 15 s) were used without and with the application of the ethanolic solution of KOH 0.2 N. It was found that the application of the ethanolic solution of KOH 0.2 N was sufficient to achieve a better BSSG extraction than without it.

#### 3.1.4. Conventional Extraction Technique: Maceration

The dried plant material was macerated in ethanol for 48 h. After that, the ethanolic extract was concentrated at low temperature and reduced pressure on a rotary evaporator (Heildolph Laborota Model 4000, Schwabach, Germany).

### 3.2. High-Performance Thin-Layer Chromatographic (HPTLC) Method

#### 3.2.1. Reagents

The reference standard used was the β-sitosterol β-D-glucoside purchased from Sigma Aldrich Co. (St. Louis, MO, USA). The HPTLC silica gel glass plate 60 F254 (10 cm × 20 cm with 250 µm thickness) was supplied by Merck (Darmstadt, Germany).

The mobile phase employed consisted of toluene, ethyl acetate and formic acid (88%). Methanol was used as a diluent for the standard solution and experimental samples.

#### 3.2.2. Standard Solution

The stock solution of BSSG (from Sigma Aldrich Co., St Louis, MO, USA) of 250.00 μg/mL was dissolved in methanol.

#### 3.2.3. Preparation of Samples

For the preparation of experimental samples, 5 mg of extracts from the MAE (5, 10, and 15 s) and the maceration extraction was dissolved in 1 mL of methanol.

#### 3.2.4. Application

To differentiate and compare each of the MAE and maceration extraction methods, each of the ethanolic extracts of *A. angustifolia* was analyzed by HPTLC for the identification and quantification of BSSG. The sample solutions were sprayed onto HPTLC plates in the form of bands of 6 mm using a CAMAG Linomat V, (Muttenz, Switzerland) with nitrogen gas and a microlite syringe of 100 mL. These samples were applied on HPTLC plates with the following properties: 20:10 cm, distance between bands, 10 mm, distance from lower edge, 8 mm, and distance from the left side, 15 mm; and 18 tracks were applied. For the quantification of BSSG, a calibration curve was used for standard solution, which consisted of six different volumes (0.5, 1.0, 2.0, 2.5, 3.0, and 4.0 μL).

For BSSG determination, 2 μL of each extract obtained by MAE at different extraction times (5, 10, and 15 s) and 4.0 μL of maceration extract (48 h) were applied three times on each plate with a constant application rate of 150 nL/s.

#### 3.2.5. Chromatography

The chromatography was performed on 20 ×10 cm 60 F254 silica gel and the development was performed in the CAMAG Automatic Developing Chamber 2 (ADC 2), Muttenz, Switzerland at 26 °C and a relative air humidity of 39%. Evaluation of several combinations of mobile phases (hexane- hexane:acetone 80:20; hexane:ethyl acetate 70:30; chloroform:methanol 90:10, toluene:ethyl acetate:formic acid 5:5:0.5) was performed. The selected mobile phase consisted of a mixture of toluene, ethyl acetate, and formic acid (5:5:0.7 *v/v/v*). The migration distance was 70 mm and the migration time was 22 min. After the development, the plate was dried with air for 5 min.

After the development of the HPTLC plate, considering that BSSG showed no sensitivity in the UV detector, the plate was sprayed with Komarovsky reagent, and after that, it was heated in the CAMAG TLC Plate Heater III, Muttenz, Switzerland at 110 °C, for 3 min.

For the evaluation of the plate images, a CAMAG TLC Visualizer Documentation System (Muttenz, Switzerland) was used, employing a high-performance 12-bit CCD digital camera with outstanding linearity. All the images were captured with an exposure time of 2 min under a UV wavelength of λ 540 nm, in a reflected way. The obtained data were processed using VisionCATS 2.4 software.

Densitometric evaluation was performed by a CAMAG TLC Scanner 4 (Muttenz, Switzerland). Absorption measurement of the derivatized plate with the Komarovsky reagent was recorded under UV light at λ 540 nm.

### 3.3. Fourier-Transform Infrared (FT-IR) Spectroscopic

The functional groups of BSSG were identified using Fourier-transform infrared (FT-IR) with a Shimadzu spectrophotometer model IRAffinity-1S (Shimadzu Corporation, Kyoto, Japan), with an attenuated total reflection accessory, which permits an analysis of samples in the medium infrared (MIR).

### 3.4. HPLC-ESI-MS

The mass spectroscopy was performed on a Bruker micrOTOF-Q mass spectrometer (Bremen, Germany) with electrospray ionization with a scanning ratio from 50 to 3000 *m/z* and a set capillary of 4500 V. The masses were processed using Bruker Compass Data Analysis 4.1 software.

### 3.5. Statistical Analysis

The data obtained were analyzed using one-way ANOVA. The Holm−Sidak Test was applied to determine significant differences between each group at (*p* < 0.001). Statistical analysis was carried out with a SigmaPlot 11.0 program [50].

## 4. Conclusions

In this work, we obtained a greater yield of BSSG from “piña” of *A. angustifolia* plants with the MAE method with the addition of KOH as a catalyst, compared to the traditional maceration method. The MAE method reduced the extraction time using ethanol as an environmentally friendly solvent. A greater concentration of BSSG was obtained with less degradation of this compound. All these advantages demonstrated that MAE was a better method to obtain BSSG from *A. angustifolia* by solid-liquid extraction than the maceration method.

## Figures and Tables

**Figure 1 molecules-24-03926-f001:**
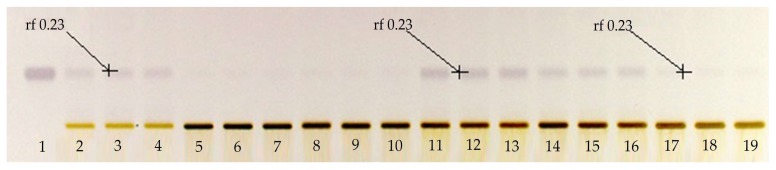
Qualitative high-performance thin layer chromatographic (HPTLC) plate derivatized with Komarovsky reagent; Track 1 standard of β-sitosterol β-d-glucoside (BSSG), Tracks 2–4 ethanolic extracts by microwave-assisted extraction (MAE) in extraction time of 5 s, Tracks 5–7 ethanolic extracts by MAE in extraction time of 10 s, Tracks 8–10 ethanolic extracts by MAE in extraction time of 15 s, Tracks 11–13 ethanolic extracts by MAE in extraction time of 5 s with KOH solution, Tracks 14–16 ethanolic extracts by MAE in extraction time of 10 s with KOH solution, Tracks 17–18 ethanolic extracts by MAE in extraction time of 15 s with KOH solution, and Tracks 16–18 ethanolic extracts of *A. angustifolia*, with white light and mobile phase toluene:ethyl acetate:formic acid (5:5:0.7 *v*/*v*/*v*).

**Figure 2 molecules-24-03926-f002:**
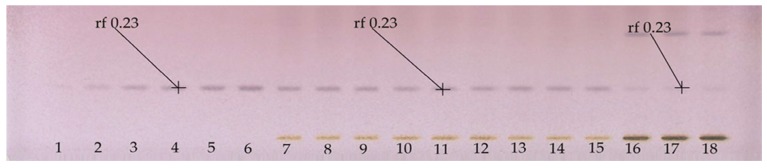
HPTLC plate derivatized with Komarovsky reagent; Tracks 1–6 calibration curve of BSSG, Tracks 7–9 ethanolic extracts by MAE in extraction time of 5 s with KOH solution, Tracks 10–12 ethanolic extracts by MAE in extraction time of 10 s with KOH solution, Track 13–15 ethanolic extracts by MAE in extraction time of 15 s with KOH solution, and Tracks 16–18 ethanolic extracts by maceration method in extraction time of 48 h of *A. angustifolia*, with white light and mobile phase toluene:ethyl acetate:formic acid (5:5:0.7 *v*/*v*/*v*).

**Figure 3 molecules-24-03926-f003:**
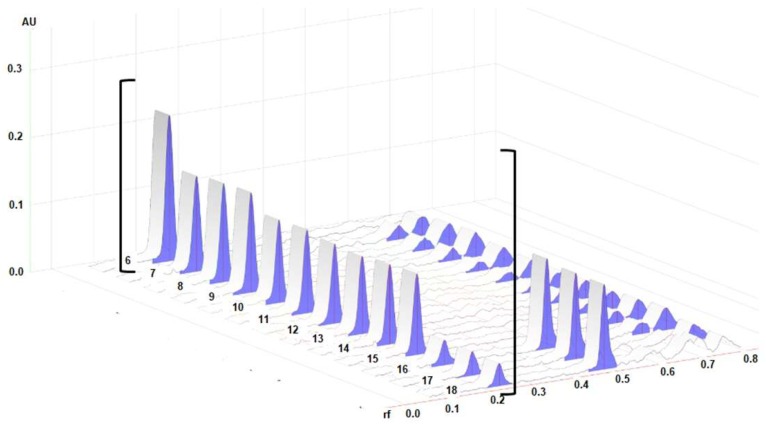
3D densitograms; track 6 represents the standard solution of BSSG, tracks 7–9 show extracts obtained by MAE in 5 s with KOH solution, tracks 10–12 are the extracts obtained by MAE in 10 s with KOH solution, tracks 13–15 show the extracts obtained by MAE in 15 s with KOH solution, and tracks 16–18 represent maceration at 48 h. All measurements were taken at λ 540 nm.

**Figure 4 molecules-24-03926-f004:**
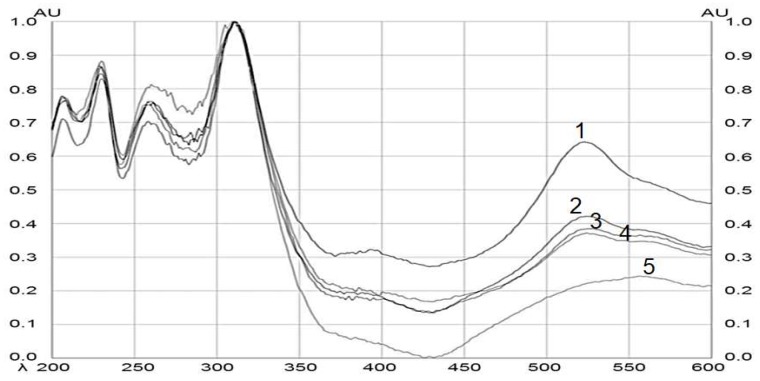
Spectral comparison of HPTLC chromatograms obtained by the BSSG standard, and ethanolic extracts obtained by MAE and by maceration of *A. angustifolia*. Wavelengths from 200 to 600 nm. (1) BSSG standard, (2) ethanolic extracts by MAE in extraction time of 5 s with KOH solution, (3) ethanolic extracts by MAE in extraction time of 10 s with KOH solution, (4) ethanolic extracts by MAE in extraction time of 15 s with KOH solution, and (5) ethanolic extracts by maceration at 48 h.

**Figure 5 molecules-24-03926-f005:**
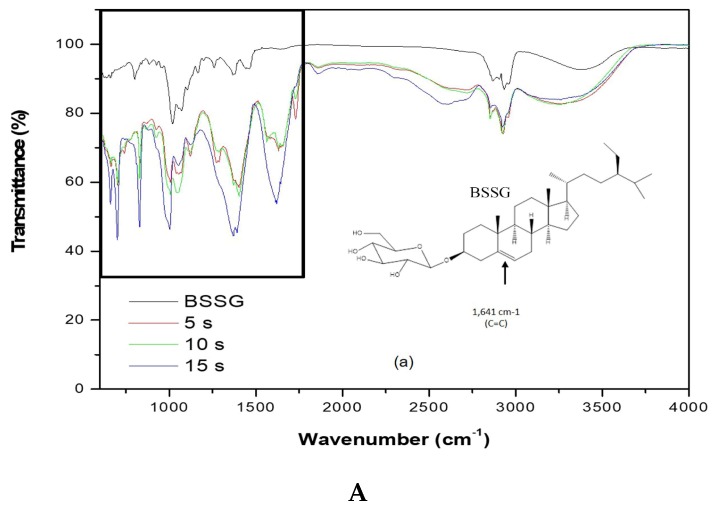
(**A**) FT-IR spectra of the BSSG standard (BSSG) and the ethanolic extracts obtained by MAE from *A. angustifolia* (5, 10, 15 s) with KOH solution; (**B**) ampliation of Figure 5A in the region from 600 to 1800 cm^−1^.

**Figure 6 molecules-24-03926-f006:**
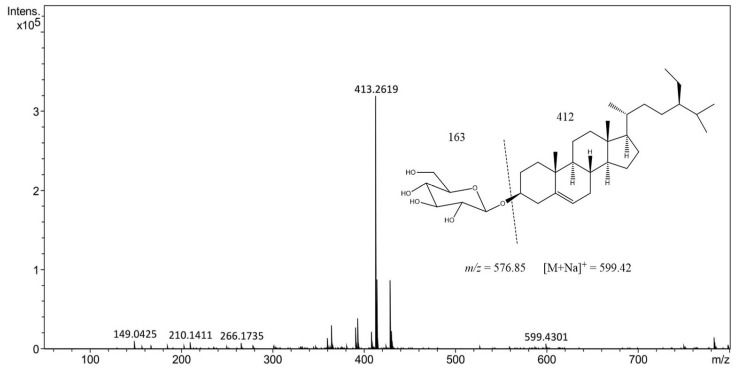
High-performance liquid chromatography−electrospray ionization−mass spectrometry (HPLC-ESI-MS) positive ion of the mass spectrum of ethanolic extract of 5 s obtained by MAE with KOH solution of *A. angustifolia*.

**Table 1 molecules-24-03926-t001:** Concentrations of BSSG with different times of MAE with KOH solution and extraction by maceration of 48 h.

Method	Extraction Time (s)	Extraction Time (h)	Mg B-Sitosterol B-d-Glucoside/Dry Weight
MAE	5	-	124.76 ± 0.0000655 ^a^
MAE	10	-	106.19 ± 0.0000793 ^b^
MAE	15	-	103.96 ± 0.0000321 ^c^
Maceration	-	48	26.66 ± 0.0000100 ^d^

Average ± SD; *n* = 3. Different letters indicate significantly different values (*p* < 0.001) according to the Holm−Sidak method. MAE: Microwave-assisted extraction.

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
