# Peer review of "Identification and Quantification of β-Sitosterol β-d-Glucoside of an Ethanolic Extract Obtained by Microwave-Assisted Extraction from *Agave angustifolia* Haw"

_molecules, 2019, doi:10.3390/molecules24213926_

Round 1

Reviewer 1 Report

After reviewing the article, I consider that the following important points should be addressed:
1. In paragraph L170-172 it is mentioned that water is an organic solvent !!! (L-171). Please rewrite this paragraph.
2. In the part where IR spectra are discussed (L-200-212), the BSSG molecule does not contain any carboxyl groups !!! Please correct this paragraph.
3. In the experimental design (L284-292), to confirm the success of the MW in the extraction process, the authors have to demonstrate with an additional experiment where KOH is not incorporated, because then the question remains if it was not the presence of this reagent which led to the rupture of the plant's BSSG molecule through a normal hydrolysis reaction and not so much by the action of the MW, since in the case of maceration, with which they are comparing their results, only ethanol was added for extraction !!!

Author Response

Response to Reviewer 1 Comments

Point 1: In paragraph L170-172 it is mentioned that water is an organic solvent!!! (L171). Please rewrite this paragraph.

Response 1Thank you for your suggestion.  We rewrite this paragraph in the text (Lines L210-211):

L210-211: ………….”.There are organic solvents, such as ethanol, methanol and acetone, which are also effective for application in the MAE”………….

Point 2: In the part where IR spectra are discussed (L-200-212), the BSSG molecule does not contain any carboxyl groups!!! Please correct this paragraph.

Response 2: Thank you for your observation. We correct this functional group for the correct group (C-OH) of secondary alcohol in the text (L-234-238) and (L-244-246).

L234-L238………….(-CH (CH3) 2) in 1366, secondary alcohol (C-OH) in 1055; while the absorption bands in 801 and 600 are due to the presence of a group -C = C-, possibly of simple glycoside. It is evident that the functional groups characteristics of BSSB were identified in the FT-IR spectrums extracts obtained by MAE 5, 10 and 15 s with KOH solution. The FT-IR spectrums also showed the presence of one C=C in the structure, characteristic of BSSG (Figure 5a) [36,37]………..

L244-L246…...It is worth mentioning that the FT-IR spectrum of extracts showed a clear displacement of some absorbances such as unconjugated olefinic (C = C) band, occurred from 1641 to cm−1, cyclic methylene groups (CH2)n band, passed from 1443 to cm−1, C-OH of secondary alcohol band……………

Point 3. In the experimental design (L284-292), to confirm the success of the MW in the extraction process, the authors have to demonstrate with an additional experiment where KOH is not incorporated, because then the question remains if it was not the presence of this reagent which led to the rupture of the plant's BSSG molecule through a normal hydrolysis reaction and not so much by the action of the MW, since in the case of maceration, with which they are comparing their results, only ethanol was added for extraction !!!

Response 3: Thanks for your suggestion. In this regard we add to the document the results of an additional experiment with which the difference between the amounts of compound extracted with and without the use of KOH as a catalyst is clearly demonstrated. This information was integrated into the document in the results section (L -98-106), in Figure 1, it was also added in the methodology (L- 332-337).

Reviewer 2 Report

The authors described an interesting method to extract β-sitosterol β-D-glucoside from Agave angustifolia Haw plant using microwaves. The method shows some advantages respect to traditional ones in terms of yield, shorter extraction times, and because it uses less toxic solvents. The method for quantification and characterization of BSSG in the extract has been described clearly and in sufficient details. I think, this article is suitable for the publication.

Author Response

Response to Reviewer 2 Comments

Point 1. The authors described an interesting method to extract β-sitosterol β-D-glucoside from Agave angustifolia Haw plant using microwaves. The method shows some advantages respect to traditional ones in terms of yield, shorter extraction times, and because it uses less toxic solvents. The method for quantification and characterization of BSSG in the extract has been described clearly and in sufficient details. I think, this article is suitable for the publication.

Response 1: No corrections were made by the referee. Thank you for your comments.

Reviewer 3 Report

The paper is reporting some founding on Identification and quantification of β-sitosterol β-D glucoside of an ethanolic extract obtained by microwave-assisted extraction from Agave angustifolia Haw. the current study includes some issues and the current version needs thoughtful revision.

The paper contains some grammar and syntax errors.

The figures (fig.3, fig4, fig 5) are of low quality and should be replaced by high-resolution illustrations.

The difference in mg β-sitosterol β-D-glucoside content between the different methods (MAE and maceration) is very high. Also, the difference between MAE time is significant. Please provide further explanation on how the 5 s MAE could make the highest extraction BSSG.

I think further investigation is required to make such a conclusion.

Based on the current founding the BSSG seems very sensitive to the experimental condition and it is rapidly degraded. This result and statement need explanation!!!? What are the potential degradation products from BSSG?

The result from figure 5 shows HPLC-ESI-MS positive ion of the mass spectrum of ethanolic extract of 5 s obtained by MAE of A. angustifolia. The chromatogram is confusing, and a BSSG standard is required to confirm this peak.

Author Response

Response to Reviewer 3 Comments

Point 1. The paper contains some grammar and syntax errors.

Response 1: Thank you for your comments. We corrected these errors.

Point 2. The figures (fig.3, fig. 4, fig. 5) are of low quality and should be replaced by high-resolution illustrations.

Response 2: We appreciate your suggestion. We replaced both figures (5 and 6) by high-resolution illustrations. Nevertheless, Figure 4 was not possible to change the lines / patterns of the lines, since it is a graphic obtained from VisionCATS 2.4 software, which consists of limited options for editing the graphic. Numbers were used for further differentiation of the lines.

Point 3. The difference in mg β-sitosterol β-D-glucoside content between the different methods (MAE and maceration) is very high. Also, the difference between MAE time is significant. Please provide further explanation on how the 5 s MAE could make the highest extraction BSSG.

I think further investigation is required to make such a conclusion.

Response 3: Thank you for your comments. We added an explanation in the text about this topic (Lines 169-190).

New Lines 169-190…….To understand these results it is important to remember the mechanism by which MAE uses microwave energy to facilitate the partition of analytes. Firstly, the rapid rise in temperature and internal pressure increase generated by the microwave radiation pushes the cell wall from inside, stretching and ultimately rupturing it, releasing of compounds. This process is caused by microwave radiation, which interacts with dipoles of polar and polarizable materials. Dipole rotation of the molecules produced by microwave electromagnetic cuts off hydrogen bonding of the cellulose, (which is the main constituent of the cell wall of the plant) which improves the migration of dissolved ions and stimulates solvent penetration into the vegetable matrix [25].

But, when we added the KOH as a catalyst, we caused more destructive effects on the vegetal matrix. This caused the BSSG to dilute and dissolve in solvent in a faster way.

In this way, we can confirm with these results that the extraction of BSSG by MAE was favored by the addition of KOH by causing a greater decomposition of the cell wall, in a short time extraction, thus avoiding the degradation of the compound when exposed to a prolonged extraction time. This was probably the reason why a greater amount of BSSG was obtained in the extraction time of 5 s with KOH solution than 10 and 15 s with KOH solution.

The scientific literature reports that one of the benefits of MAE is the greater amount of the compound of interest obtained, compared to conventional extraction methods, such as maceration. The maceration method consists of the penetration of the solvent into the cell, causing dehydration or rupture of cell membranes [21].

In this work, the large amount of BSSG obtained by MAE compared to the method of maceration, is in itself due to the same MAE technique, which also benefits from the addition of the KOH solution……

Point 4. Based on the current founding the BSSG seems very sensitive to the experimental condition and it is rapidly degraded. This result and statement need explanation!!!? What are the potential degradation products from BSSG?

Response 4: Thank for your comment. We added more information about this topic (Lines 237-246)

New Lines 251-257 ……………… It is necessary to mention that the degradation of sterols, takes place at more than 150 ° C, resulting in fragmented phytosterol molecules, volatile compounds and oligomers [38]. In this research work, the maximum extraction temperature was 23 ° C, for this reason, we believe that the reason for the possible degradation was caused by the addition of KOH solution and not by microwave energy. The scientific literature mentions that the microwaves used in the extraction of bioactive compounds are a source of heat without contact, for this reason; there is a decrease in thermal degradation [22].

Point 5. The result from figure 5 shows HPLC-ESI-MS positive ion of the mass spectrum of ethanolic extract of 5 s obtained by MAE of A. angustifolia. The                                                                                                                                                                                                                                                                                                                                                                                                                                                                                                                                                                                                                                                                                                                                                                                                                                                                                                                                                                                                                                                                                                                                                                                   chromatogram is confusing, and a BSSG standard is required to confirm this peak.

Response 5: Thank for your comment. We modify the Figure 6 with a better description in the chromatogram of the possible site of the ionization in the BSCG molecule, and we complement the text with other sentence (Lines 266-270)

New Lines 275-279 ………. Our compound of interest BSSG has a real molecular weight of 576, and the mass spectrum in the 5 s ethanolic extract obtained by MAE shows a peak at 599.42 m / z = [M +Na] +, which is due to an adduct of a Na+. According to the spectral data found in this research work, it was confirmed that the compound identified and quantified from an ethanol extract of A. angustifolia obtained by MAE of 5 s is BSSG (Figure 6) [36].……………..

Round 2

Reviewer 1 Report

The authors attended all the recommendations suggested by this reviewer.

Author Response

Thank you for your comments

Reviewer 3 Report

The current version of the manuscript was revised carefully by the authors. Before acceptance please consider the following comments:

The paper contains some grammar and syntax errors and should be revised thoroughly. Latin names in whole manuscript should be in italic: (ex. Page 8 line 259: angustifolia)

Author Response

Thank your for your comments, we changed to italic "angustifolia" Pag 8 line 259